# Motivation as a Measurable Outcome in Stroke Rehabilitation: A Systematic Review of the Literature

**DOI:** 10.3390/ijerph20054187

**Published:** 2023-02-26

**Authors:** Giulio Verrienti, Cecilia Raccagni, Ginevra Lombardozzi, Daniela De Bartolo, Marco Iosa

**Affiliations:** 1Department of Neurorehabilitation, Casa di Cura Villa Verde, 73100 Lecce, Italy; 2Department of Neurology, Provincial Hospital of Bolzano (SABES-ASDAA), Lehrkrankenhaus der Paracelsus Medizinischen Privatuniversität, 39100 Bolzano, Italy; 3Department of Neurology, Innsbruck Medical University, 6030 Innsbruck, Austria; 4Von Siebenthal Neuropsychiatric Clinic and Hospital, 00045 Rome, Italy; 5Smart Lab, IRCCS Santa Lucia Foundation, 00179 Rome, Italy; 6Department of Psychology, Sapienza University of Rome, 00185 Rome, Italy

**Keywords:** motivation, participation, apathy, stroke, neurorehabilitation, performance

## Abstract

Motivated behaviours are thought to lead to enhanced performances. In the neurorehabilitation field, motivation has been demonstrated to be a link between cognition and motor performance, therefore playing an important role upon rehabilitation outcome determining factors. While motivation-enhancing interventions have been frequently investigated, a common and reliable motivation assessment strategy has not been established yet. This review aims to systematically explore and provide a comparison among the existing motivation assessment tools concerning stroke rehabilitation. For this purpose, a literature search (PubMed and Google Scholar) was performed, using the following Medical Subject Headings terms: “assessment” OR “scale” AND “motivation” AND “stroke” AND “rehabilitation”. In all, 31 randomized clinical trials and 15 clinical trials were examined. The existing assessment tools can be grouped into two categories: the first mirroring the trade-off between patients and rehabilitation, the latter reflecting the link between patients and interventions. Furthermore, we presented assessment tools which reflect participation level or apathy, as an indirect index of motivation. In conclusion, we are left to put forth a possible common motivation assessment strategy, which might provide valuable incentive to investigate in future research.

## 1. Introduction

Although the concept of motivation may intuitively seem clear and simple, it poses some hidden pitfalls. What do we mean when we say that a certain treatment “motivates” patients? Is motivation a measurable parameter? Can we turn to motivation as it was a clinical parameter? Despite intense research about this topic, these questions remain probably open. In fact, motivation is a complex, multifaceted psychological construct, resulting from the interplay of several factors. Up to now, indirect measurements, which may provide a quantification of motivation, are available.

Motivation has been didactically defined as an orientation for which humans and other animals activate and sustain behaviour toward a goal [1]. This definition can be applied to several daily life scenarios. Motivation and its related aspects have been the subject of numerous research studies and several competing theories, concerning the content of motivated behaviours, have been regularly proposed. Among incentive theories, a distinction between extrinsic and intrinsic motivation occurs frequently [2]. While intrinsic motivation is related to the joy or to the interest by doing a certain activity, extrinsic motivation occurs when the goal of an activity is an external reward, which is separated from the activity itself. Research on motivation has been employed in multiple areas, including applications in business [3,4,5], educational [6,7], and wellness fields [8]. A specific branch of medical research focused on determining factors which improve recovery from diseases. In this context, motivation is of particular interest to neurologists, neuropsychologists, and experts in neurorehabilitation since it is closely related to therapy outcome. Not surprisingly, highly motivated patients are prone to reach a better recovery than low motivated patients [9]. In the field of clinical research, a common measurement strategy of motivation represents an unmet need.

This review aims at first to illustrate an overview of the most important general motivation theories and, second, to place them with regard to the stroke rehabilitation literature. Moreover, we investigated and compared motivation assessments, which have been used in the last years in interventional studies on stroke patients in the rehabilitative setting. Finally, at the light of the literature, we discussed how to assess motivation, which might be helpful in further clinical research.

### 1.1. Theories on Motivation: An Overview

An academic, valid distinction between process and content theories about motivation has been established: while process theories try to explain how and why motivation influences behaviours, content theories attempt to define needs that motivate peoples’ actions. Among process theories, self-determination theory (SDT), social-cognitive theory (SCT), and goal orientation theory (GOT) have been deeply investigated. All the above-mentioned theories have been developed in different fields, in particular in the learning-field. Some of these (especially the SDT, but also GOT and SCT) have been further extended to re-learning processes, applying them in the neurorehabilitation setting. SDT represents a macro theory of human motivation [3]. The concept of SD refers to the person’s faculty to make choices, having a great impact on motivation. In fact, people feel more motivated to act (“to make something”) if they observe an effect on the outcome of their own action. According to Deci and Ryan [10], people tend to be driven in their actions by a need to grow and gain fulfilment. In the cognitive evaluation theory, a sub-theory of SDT, a subject becomes self-determined when needs for competence, connection, and autonomy are satisfied. While autonomy refers to the human need to feel in control of behaviours, competence means feeling confident with different task-related skills. Consequently, a person who is confident to reach a goal is also more prone to take actions. Connection is linked to the concept of relatedness. In this sense, a person needs to feel a sense of belonging to the community. In the SDT, three main motivation domains were identified: amotivation (lack of motivation), extrinsic motivation, and intrinsic motivation. While in the intrinsic motivation an action is the result of an intrinsic will regulation, i.e., a person makes something only for the enjoyment of the action, amotivation results in inaction or action without real will. Inside extrinsic motivation many different levels of regulation are related to different types of external value and to action’s consequences (punishments/rewards).

In the SCT, motivation is the result of a cognitive process which is influenced by personal, behavioural, and environmental factors. An essential role in this theory is played by the subject’s faculty to change and manipulate the environment to reach personal goals. This faculty is strongly influenced by the subject’s beliefs about their own capabilities. The most important belief in the SCT is the (perceived) self-efficacy, defined by Albert Bandura as “people’s judgment of their capabilities to organize and execute courses of action required to attain designated types of performances. It is concerned (…) with judgments of what one can do with whatever skills one possesses” [11].

The term goal orientation (GO) refers to the cognition of the achievement and its implications on the behaviour responses [12]. GO influences the individual’s cognitive or emotional tendency toward events, which in turn will trigger behavioural responses. Within GO theories, orientation to achieve goals differs individually and it is related to a subconscious subject’s predisposition. While some subjects show motivated behaviours in task completing exclusively to align themselves with the community’s expectancy (performance goal orientation), a mastery goals orientation leads to motivated behaviours, principally through intrinsic values of related actions. The most important difference with performance goal orientation consists in the conviction that the required ability for task completing can improve through skills training. A summary of these theories is reported in Table 1. In the next paragraph, it will be presented how the most prominent aspects of these theories have been applied in the field of neurorehabilitation.

### 1.2. Motivation in Neurorehabilitation

Several descriptive studies and reviews describe stroke as a sudden interruption of what was otherwise expected to be a normal life [18,19]. According to the WHO definition [20], rehabilitation in patients with disabilities should be considered as the “process aimed to reach and maintain optimal physical, sensory, intellectual, psychological and social functional levels”. With reference to the rehabilitative setting, motivation may be defined as the reason why patients activate sustained efforts toward the recovery. However, a gap exists in the rehabilitation literature about the nature of motivation. Maclean and Pound [9] divided the global literature about this topic into three broad groups: the first group (the prevalent one) describes motivation as a purely internal quality of the patient; the second group indicates motivation mainly as a social driven factor; and the last group sees motivation as a combination of social factors and individual clinical characteristics, such as personality. A first pitfall about the psychological conceptualization of motivation in rehabilitation could be the differentiation between internal and external motivation. From a mere cognitive point of view, the achievement of a motor functional improvement could be seen as an external reward factor motivating the patient, but from a behavioural point of view, it is an intrinsic factor because it is internal to the patient/person. This aspect has been deeply treated in some studies focused on patients with stroke, for whom motor and cognitive deficits are strictly intertwined.

#### Focus on Motivation in Stroke

Most of the studies regarding motivation in neurorehabilitation are focused on stroke. Among these studies, many researchers turn to motivation as a process of the SDT [21]. Focusing on the role played by both intrinsic and extrinsic motivation, SDT longs to provide a comprehensive explanation of internal and external influences on human behaviours. In particular, as proposed by Yoshida et al. [22], motivation in subacute stroke patients is mainly influenced by extrinsic reward factors (e.g., positive feedback provided by therapists, praises by medical staff or relatives, etc.). Then, if the self-regulation is, also partially, reached through the satisfaction of basic psychological needs (autonomy, competence, and relatedness), one’s motivation moves from an extrinsic level to an intrinsic one [23]. Therefore, in case of a valid functional improvement, the patient may attend the rehabilitation almost exclusively supported by intrinsic motivation. In addition, the importance of internalization of regulations in fostering rehabilitation and physical activity has been demonstrated, indicating that an enhanced competence for exercise is a positive predictive of more adaptive exercise behavioural outcomes [24]. Moreover, the process of internalization of behavioural regulations has been shown to be the basis for long-term improvement after conclusion of treatment [25]. According to SDT, motivation, especially in stroke rehabilitation, is a dynamic phenomenon. If the purpose of tailoring interventions to extrinsic factors is essential in maintaining patients’ motivation (at least in the subacute phase) [22], clinicians should promote anyways the process of internalization. In this context, sharing an appropriate amount of information about rehabilitation and avoiding “mixed messages” [26] are useful strategies. However, it should be finally noted that both intrinsic and extrinsic motivation are important in fostering actions: the relationship between the two motivation types is not conflictual but addictive, as suggested by Cerasoli et al. [27].

According to the original works of Locke and Latham [12,16], a recent, comprehensive review [28] on GOT identified a triad of different goal types with regard to neurorehabilitation. These are represented by performance goals (where patients use previously learned strategies to perform tasks), process-oriented or learning goals (where required skills for improving performance have to be learned), and outcome goals (which are related to the performance of competitors; this type of goal has not been investigated so far, but a deep research study about this topic is in progress, e.g., [29,30]). The goal setting should represent a milestone of the neurorehabilitation setting [31], although no single guideline has been established [32]. In fact, a recent review [33] identified at least 12 different approaches to goal setting. Among these, two common features have been identified: having measurable goals and patients’ involvement in goal setting. Within the GOT literature, a ‘patient-centered’ approach to establish goals for rehabilitation is universally accepted. Although evidence to support its efficacy is weak [34], a patient-centered approach is supposed to improve self-perceived participation, performance itself, and functional outcomes [35,36] in stroke patients. Furthermore, GO instructions may result in significant increases in the rate of repetitions of exercise in stroke rehabilitation [37].

With regard to SCT principles, personal and environmental factors reciprocally interact influencing motivated behaviour: in this sense, the individual exerts an influence on the environment, which, in turn, influences him/her. A high level of self-efficacy is directly related to the quality of life in stroke patients [38] and to the amount of daily physical activity [39]. Stewart et al. [40] found self-efficacy to be a significant predictor of the performance of the affected arm in chronic stroke patients with mild motor impairment. The authors concluded that self-efficacy may serve as a target for interventions to improve proximal arm control after stroke. Another study [41] demonstrated that self-efficacy for walking, and in general, to perform physical exercise, predicts higher exercise adherences in individuals with chronic stroke.

## 2. Materials and Methods

A systematic review was performed according to Preferred Reporting Items for Systematic Reviews and Meta-Analyses (PRISMA) guidelines. A literature search (PubMed and Google Scholar) was performed on 22 August 2022, and we selected articles published in the timeframe 1 January 2010 to 30 Juny 2022, using the following Medical Subject Headings (MeSH) terms: “Assessment” OR “Scale” AND “motivation” AND “stroke” AND “rehabilitation”. In addition, a backward search (checking the bibliography of identified papers) was conducted to identify any studies that were not retrieved using the main search strategy. The inclusion criteria were (1) articles published in English language; (2) interventional studies in stroke patients in rehabilitation setting, which examine the direct effect of the intervention on motivation, measured by specifically designed assessment tools; (3) original research examining the indirect effect of an intervention on motivation (in this case, the motivating effect of the intervention is measured by indirect motivation assessment tools, such those specifically designed to assess participation or depression); and (4) studies conducted in the above-reported period. In the main search strategy, the following exclusion criteria were adopted: (1) article types such as letters to the editor, case reports, reviews, and meta-analyses; (2) studies for which the complete text could not be found; and (3) articles not in English.

Two authors independently performed all searches and removed duplicate records. Finally, a third author performed a quality assessment.

### 2.1. Quality Assessment

To assess the quality of our review, we have used a qualitative approach. This approach is reasonable because we are investigating phenomena (motivation assessment), in which the assessment’s choice in the examined studies was arbitrary. Thus, we used the qualitative methodological checklist of the National Institute of Clinical Nursing (NICE) [42]. According to the NICE checklist, ++ means that all or most of the checklist criteria have been fulfilled, + means that some of the checklist criteria have been fulfilled, and—means that few or no checklist criteria have been fulfilled.

### 2.2. Data Extraction

The data extraction form developed for this study aimed to carry out study characteristics (title, first author, study design, geographical location, study setting, and aim of the study with description of the adopted motivation assessment tool, including frequency of assessment). Two authors, who conducted the study selection independently, performed the data extraction. If necessary, any disagreements were discussed with a third review author.

### 2.3. Ethics Statement

This study is a systematic review and does not deal with human participants.

## 3. Results

Electronic and additional sources identified 330 references. Duplicate articles were removed, leaving a total of 324 articles. A total of 278 of these were excluded after assessing their abstract, as they did not meet the inclusion criteria. Another study [43] was removed after reading, because, although it has met inclusion criteria, the trial was not completed at the time of writing this article. Thus, a total of 46 articles [44,45,46,47,48,49,50,51,52,53,54,55,56,57,58,59,60,61,62,63,64,65,66,67,68,69,70,71,72,73,74,75,76,77,78,79,80,81,82,83,84,85,86,87,88,89] published between 2011 and 2022 were assessed for eligibility after reading the full text (Figure 1) and underwent the quality control.

The 46 selected studies were published between 2010 and 2022. The review includes 31 RCTs and 15 CTs. Of the 46 studies, 25 were carried out in Europe, 12 in Asia, 8 in America (7 in the USA and 1 in Brazil), and 1 in Australia (Table 2).

Most of the trials are dealing with robotic therapy, exergames, and virtual reality. In the remaining studies, different types of interventions such as feedback-mediated exercises; occupational therapy; specific physiotherapy approaches; integrated social and educational approach; music, dance, and pet therapy; and competitive rehabilitative strategies are included. Interestingly, we have not found any study regarding drugs, such as antidepressants or stimulants, and effect on motivation in this patient target. The setting in which the intervention was applied was in most cases the chronic phase of stroke (>6 months from stroke onset). The Intrinsic Motivation Inventory [90] represents the most-used motivation assessment tool in our review (see Figure 2). The frequency of motivation assessments appears to be extremely variable in the different studies. The quality control, performed according to the NICE guidelines, is reported in Appendix A (Table A1).

## 4. Discussion

There are many ways to define motivation for rehabilitation. The reason for this is rather simple: the concept of motivation is in practice poorly understood and, as suggested by Maclean et al., also the adopted criteria among professionals to recognize motivation have been shown to have blurred boundaries [91]. In this context, one possible approach in motivation’s definition refers to the orientation for which patients activate sustained efforts toward the recovery [22]. Moreover, the idea of motivation might be also referred to the patient’s self-involvement within the rehabilitation context (rehabilitation-motivation). It should be noted that a motivation questionnaire which investigates how motivation is influenced by a treatment (e.g., the Intrinsic Motivation Inventory) does not quantify the global motivation for rehabilitation per se, but almost exclusively the subject’s predisposition/inclination to the specific treatment. In the different types of studies, it is often implied which meaning of motivation assessment (rehabilitation or intervention motivation) is intended, but such for a dichotomy has not previously been explicit in the literature we reviewed. Furthermore, in some studies of our review, motivation has been also indirectly quantified by depression or participation-related assessments [61,65,69,76,79,80,85,88].

### 4.1. Motivation Assessment Scales in Stroke-Rehabilitation

To our knowledge, three patient-rated scales, specifically designed for stroke patients in rehabilitation, are available: the Stroke Rehabilitation Motivation Scale (SRMS) [92], the Motivation for Rehabilitation scale (MORE scale) [93] and the Adapted Achievement Motivation Questionnaire (AAMQ) [94]. Another questionnaire to assess motivation in stroke survivors was developed by Hallams and Baker [95], but the questionnaire’s reliability could not be determined because of the small size of the examined sample. The MORE scale, recently developed by Yoshida et al. [93], is a promising motivation assessment tool in stroke patients. It consists of 17 items and explores both extrinsic and intrinsic motivation by referring to 2 types of factors (personal and social-relationship factors). The Cronbach’s alpha coefficient, evaluated for assessing internal consistency, was excellent (0.948). With regard to test–retest reliability, a moderate correlation was found between scores at the beginning and one month after hospitalization (rho = 0.612. *p* < 0.001). Furthermore, a strength of this tool is its relative independence from some confounding factors, e.g., depression or apathy. The SRMS [92], based on the SDT, has been adapted to be suitable among stroke patients from the Sports Motivation Scale [96], a motivation assessment tool used in sports. The SRMS consists of 28 items exploring 3 domains (amotivation, extrinsic motivation, and intrinsic motivation). A shorter 7-item SRMS version was proved to have good reliability among the rest of the 28 items. In the 28-item version, motivation is quantified through 7 subscales (extrinsic motivation (EM)-introjected; EM-regulation; EM-identification; amotivation; intrinsic motivation (IM)-knowledge; IM-stimulation; and IM-accomplishment). Each subscale includes more questions. A major limitation of the original version of the SRMS consists in the absence of a comparison with similar scales at the time of its validation, due to the fact that none existed yet. However, its validity has been indirectly proved in translation studies in other languages [97], whereby the K-SRMS includes only 24 items of the original SRMS. The first difference between the SRMS and the MORE consists in the item formulations, which are questions in the SRMS and statements in the MORE. Regardless of this psychometric difference, in some cases, the MORE and the SRMS share items or items have the same contents (MORE items: #5, #4, respectively, with SRMS items: #12, #8). In other cases, items of both scales are similar (MORE items: #6, #17, #9, respectively, with SRMS items: #24, #21 and #1, #18). The categories “goal setting”, “influence from supporters”, and “resilience against obstacles”, originally identified in the MORE scale, are well represented also in the SRMS. The category “success experience”, which is related to stroke patients’ functional improvement [98,99], is better represented in the MORE scale (items #11, #12, 13#) than in the SRMS. The intrinsic psychological effect of rehabilitation is conversely more investigated by the SRMS (items #5, #6, #15, #20). Finally, as mentioned above, the MORE seems to be not influenced by depressive symptoms, while the SRMS did not show predictive validity with mood-relevant measurements (such as anxiety, depression, and stress, as measured by Park et al. [97]). Finally, the AAMQ [94] consists of 28 items, shows an interesting internal consistency (Cronbach’s alpha coefficient: 0.946), and was regionally validated for examining the motivational level of Iranian stroke survivors. The content validity was approved by an expert panel. It was not possible to determine neither the criterion validity (there was no regionally standardized measure for comparison at the time of the pilot study) nor the construct validity (a factorial analysis could not be performed due to the small sample size). The main characteristics of the three above-reported scales are summarized in Table 3.

In the stroke rehabilitation literature, many other motivation assessment tools have been used. In addition, several interventional studies provided a motivation assessment, which is primarily based on self-report scales and questionnaires. In our review, the Intrinsic Motivation Inventory (IMI) [90] is the most used motivation assessment tool. The IMI aims to measure the levels of intrinsic motivation as the outcome of a set of subscales (interest/enjoyment, perceived competence, effort, value/usefulness, pressure/tension, relatedness, and perceived choice) and shows, in the original version, a good internal consistency with a Cronbach’s alpha coefficient of 0.85 [90]. The original scale consists of 45 items. The authors of the IMI have suggested that different sets of the original scale can be used, depending on subscales’ relevance to the issues researchers are exploring. Items can also be removed if they “sound” redundant. In the overall inventory, the interest/enjoyment subscale is considered the most direct self-report measure of intrinsic motivation. The Vitality Index (VI) [100] is an easy to administrate, medical staff-rated scale to assess vitality with regard to activities of daily living. It consists of 5 items (waking up, communication, feeding, on and off toilet, and rehabilitation/activity) and all items are answered with a 3-point scale. The presence of item #5 (rehabilitation/activity) in its framework justifies the use of VI as an indirect motivation measurement instrument [76]. The reliability of the VI was examined by determining test–retest reliability (0.98), interrater reliability (0.14), and internal consistency (Cronbach α: 0.88). In other studies of our review, scales concerning the motivation for physical activity have been used. The Echelle de Motivation envers l’Activité Physique (EMAP) [101] consists of 18 items, which cover the 6 forms of motivation underlined by SDT, and can be considered as a valid, reliable, and patient-rated tool to assess motivation in the rehabilitation context. In a validation study of the EMAP in the Spanish population [102], the factor analysis confirmed the original six dimensions of EMAP. A validation of the EMAP in the English language has not yet been performed. Finally, the Behavioural Regulation in Exercise Questionnaire [BREQ], proposed in the original version by Mullan et al. [103], was a first attempt to develop an instrument able to quantify the behavioural regulation in the exercise domain. The BREQ-3 [104,105], is a patient-rated scale which investigates the reasons underlying peoples’ decisions to engage in physical exercise. It consists of 24 items with a 5-point Likert scale and includes six subscales, assessing amotivation, external, introjected, identified, integrated regulation, and intrinsic motivation. Its validity was confirmed by several validation studies in other languages and, recently, it has been proved to be invariant across very different age groups [106].

### 4.2. Motivation, Depressive Features and Participation in Stroke Patients

Motivational deficits are prevalent in several psychiatric disorders including depression, where a persistent lack of motivation is a pivotal symptom. In the post-stroke depression (PSD), which occurs in approximately one in three stroke-surviving patients [107], motivational impairments have been previously described. The importance of diagnosis and treatment of this disease is crucial because of its negative impact on the rehabilitation outcome [108]. In fact, as shown in a retrospective, case-control study, PSD was demonstrated to be an additional disabling factor which is responsible for ~15% of the increased disability [109]. In a factor analysis of the PSDRS [110], an assessment scale specifically developed for PSD, Quaranta et al. [111] identified a factor, named “Reduced Motivation”, resulting from the combination of apathy and anhedonia. Anhedonia, defined as an inability to experience pleasure, is one of the core symptoms of major depressive disorder (MDD) [112] and, as feature of PSD, has been associated with increased levels of depression at hospital discharge [113]. Anhedonia has been related to cognitive deficits, including executive dysfunction [114], and has been associated with the disruption of neural circuits and neuroendocrine impairments in stroke patients [115]. Although anhedonia is well known to play a relevant role in motivational aspects of (selected) post-stroke patients, very few studies concerning its role in rehabilitation is available, while many studies investigated the role of apathy. Apathy is defined as a lack of motivation relative to the patient’s previous level of functioning, accompanied by a quantitative reduction of goal-directed behaviour and cognition [116]. Starkstein et al. [117] found that 18 of 80 consecutive patients (22.5%) with an acute cerebrovascular lesion met this criterion for apathy. In addition, Santa et al. [118] observed that 14 patients (21%) among 67 subacute stroke subjects were apathetic. As expected, these patients, if compared with non-apathetic patients, showed less improvement in functional independence. Although often associated with depression and cognitive impairment, apathy may occur independently of both [119], but may often be misdiagnosed as depression due to symptom overlap [120]. In general, it is well known that cognitive impairments in depression are strictly connected to impairments in motivational processes [121]. As an example, Kanellopoulos et al. found in a sample of 135 older adults (age ≥ 50) with PSD that apathy is the only symptom cluster of PSD with a significant relationship to cognitive impairment across several neuropsychological domains [122].

In the literature, several tools to assess PSD are available. In addition to the above-mentioned PSDRS [110], the Beck’s Depression Inventory (BDI-II) [123], the Hamilton Depression Rating Scale (HAM-D) [124], the Montgomery-Åsberg Depression Rating Scale (MADRS) [125], and the Zung Self-Rating Depression Scale (ZSRDS) [126] are widely used, providing an adequate measure of depression in stroke patients [127]. Anyways, it should be noted that a PSD assessment in toto may provide an indirect motivation assessment. In fact, these scales are designed to investigate all aspects of depression and do not directly address the motivational features. In our review, some studies have used valid PSD assessment tools, such as the 10-item Center for the Epidemiological Studies of Depression Short Form (CES-D10) [128], the Apathy Evaluation Scale (AES) [129], and the Geriatric Depression Scale—15 items (GDS-15) [130]. Among these tools, we believe that AES, because of its intrinsic nature focused on apathy, could probably be considered as the most suitable motivation assessment tool. This consideration is also supported from other studies concerning this scale and success in rehabilitation [131,132].

With regard to learning theories, participation can be defined as the degree of active involvement of a student in classroom learning activities. Similarly, we consider participation in the neurorehabilitation setting as the process that drives patients to be an active part of the decisions and of the activities that influence their recovery. Engagement in rehabilitation and active involvement in the rehabilitation program are in fact crucial for recovery. A qualitative study using semi-structured interviews [26] found differences among 22 stroke patients, identified by professionals as having either “high” or “low” motivation in rehabilitation. In contrast to low motivation patients, high motivation patients were more involved, more likely to understand the purpose of the neurorehabilitation, and more aligned to the aims and methods of rehabilitation professionals.

Motivation and participation are strictly linked, e.g., motivational deficits trigger unavoidable consequences at the participation level [133]. In this sense, participation assessment can be considered as a surrogate to assess the motivation level. Based on the rating of medical staff, the Pittsburgh Rehabilitation Participation Scale (PRPS) [134,135] evaluates indirectly the patients’ motivation. It consists of a single item. The examinator rates, on a scale from 1 to 6, the patient’s participation (effort and motivation as perceived by the examinator) in the therapy session. As it is based on the clinicians’ observation without any patient self-report, the PRPS might not reflect the patients’ real motivation. The revised Patient Participation in Rehabilitation Questionnaire (PPRQ) [136] is a 20-item questionnaire, developed for patients with various neurological diseases (including stroke), which rates patients’ experiences of participation in neurological rehabilitation. This scale evaluates five participation aspects, including the motivation domain. Finally, the Participation Scale (P-scale) [137] is a participation assessment tool, developed by an international team, which measures (social) participation for use in rehabilitation, stigma reduction, and social integration programs. In our review, the P-scale was used by [79].

## 5. Limitations

The findings of this review are limited by inclusion dates. We chose to report on tools used in the past 12 years, due to the significant progress of the concept of patient motivation in rehabilitation over the last decade. For example, the two most promising assessment tools (SMRS and MORE), specifically designed for stroke patients in rehabilitation, were developed, respectively, in 2012 and 2022. We expect that other assessment tools, eventually characterized by the presence of some “adjustment factors” (i.e., for patients’ age, for initial impairment severity, etc.), will be developed in the future.

## 6. Conclusions and Recommendations for Future Studies

Due to its role in rehabilitation as a determining factor of outcome, a reliable validated instrument to measure motivation is paramount. As highlighted in this review, in the stroke literature, several motivation assessments tools have been developed. These are grouped into two categories: the first including tools exploring the relationship between patient and rehabilitation and the latter reflecting the trade-off between patient and intervention. An indirect estimation of the motivation level can be also obtained through assessment tools specifically designed to measure participation or apathy. We believe that the simultaneous administration of combined patient- and medical staff-rated motivation assessments may provide an overall measurement tool of motivation and reduce bias. Finally, motivation represents a dynamic phenomenon, whose level can be enhanced or diminished, sometimes in short timelapses, by several factors. For this reason, motivation should be regularly assessed, especially in rehabilitation, where the clinical course may be long lasting and prone to multiple changes. A frequent assessment might be helpful to identify possible motivation decreases and, potentially, to react appropriately by adapting therapeutic strategies.

## Figures and Tables

**Figure 1 ijerph-20-04187-f001:**
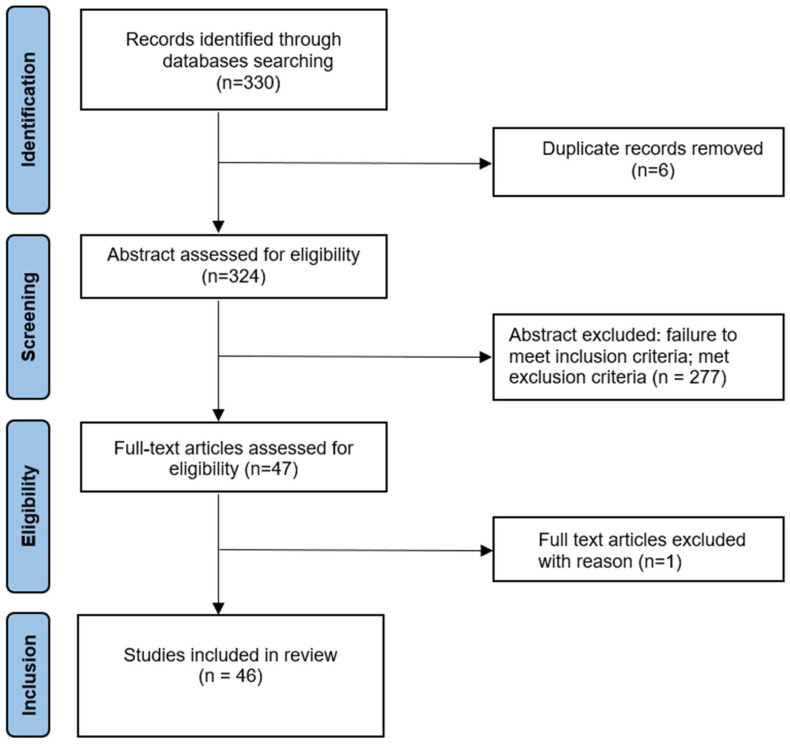
Flow diagram of study inclusion.

**Figure 2 ijerph-20-04187-f002:**
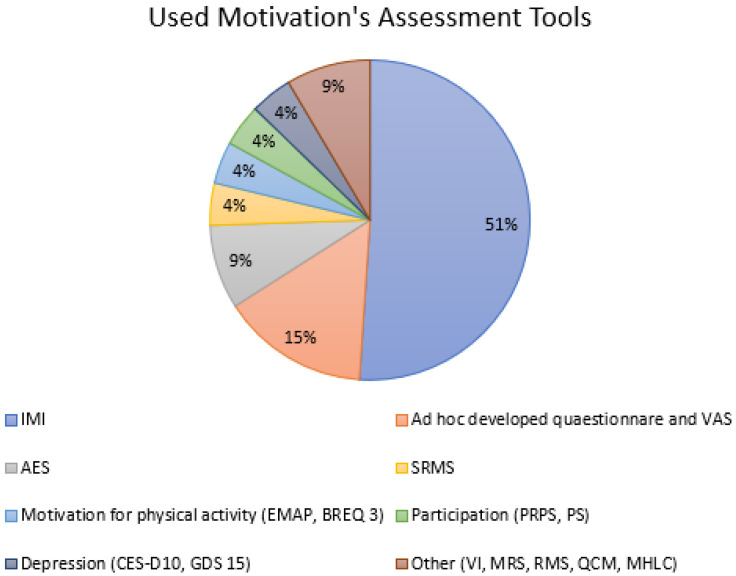
Used motivation assessment tools. Abbreviations: IMI: Intrinsic Motivation Inventory; AES: Apathy Evaluation Scale; MHLC: Multidimensional Health Locus of Control scale; CES-D10: 10-item Center for the Epidemiological Studies of Depression Short Form; BREQ-3: Behavioural Regulation in Exercise Questionnaire; SMSC: Sport- and Movement-Specific Self-Concordance Scale; GDS-15: Geriatric Depression Scale—15 items; MAPS: echelle de Motivation envers l’Activité Physique en context de Santé; SRMS: Stroke Rehabilitation Motivation Scale; QCM: Questionnaire for Current Motivation; VI: Vitality Index; RMS: Rehabilitation Motivation Scale; PRPS: Pittsburgh Rehabilitation Participation Scale; PS: Participation Scale; MRS: Motivation of Rehabilitation Scale; VAS: Visual Analogic Scale.

**Table 1 ijerph-20-04187-t001:** Summary of contemporary motivation theories.

	SDT	SCT	GOT
Main Concept	People are driven in their actions by a need to grow and gain fulfilment. SDT is focused on the role played by both intrinsic and extrinsic motivation. While intrinsic motivation is related to the interest by doing a certain activity, extrinsic motivation occurs when the goal of an activity is an external reward. Intrinsic and internalized motivations are promoted by feelings of competence, autonomy, and relatedness.	Motivation is the result of a cognitive process which is influenced by personal, behavioural, and environmental factors. The subject’s judgment of their own capabilities to reach a specified level in the performance (Self-efficacy) is the main driver of motivated action.	The term goal orientation (GO) refers to the cognition of the achievement on the behaviour responses. Achievement goals are defined as the terminal point towards which one’s efforts are directed. Two different GOs have been described: In the performance GO (PGO), the subjects show motivated behaviours only to align him/herself with the community’s expectancy. In the mastery GO (MGO), motivated behaviours are characterized by intrinsic values of related actions. In the MGO, the required ability for task completing is supposed to improve through skills training. In PGO, the required task’s ability is a congenital fixed trait (entity mindset): in this case, the real motivation in task’s completing consists in showing to the community that the one is enough able to
Keywords	Competence, autonomy, relatedness	Self-efficacy	Mastery goal orientation, performance goal orientation
Main references	Deci and Ryan [2,3,13,14]	Bandura [11,15]	Locke and Latham [12,16]; Dweck [17]

Abbreviations: SDT: self-determination theory; SCT: social-cognitive theory; GOT: goal orientation theory; MGO: mastery goal orientation; PGO: performance goal orientation.

**Table 2 ijerph-20-04187-t002:** Motivation-assessing interventional studies in stroke patients in rehabilitation setting.

First Author	Year	Study Design	Study Location	Stroke Setting	Aim of the Study	Motivation Assessment Tool	Assessment Frequency
Bergmann J [44]	2018	RCT	Germany	SA	To evaluate the acceptability of robot-assisted gait training with and without virtual reality	IMI	After the 1st, 6th, and 12th week and after the crossover therapy session
Winter C [45]	2021	RCT	Germany	C	To evaluate the acceptability of robot-assisted gait training in each of three different experimental conditions (VR via HMD, VR via monitor, and treadmill training without VR)	IMI	After every condition
Guillén-Climent S [46]	2021	CT	Spain	DS	To assess the usability of a robotic device combined with a telecare platform, in which the training is based on serious games for upper limb rehabilitation in the home environment	IMI	On the last day of treatment
Navarro MD [47]	2020	RCT	Spain	C	To investigate the effectiveness and motivation of a group-based intervention, combining conventional and computerized multi-touch exercises, when administered in either a competitive or non-competitive manner	IMI	Before and after each intervention
Swinnen E [48]	2017	CT	Belgium	C	To examine stroke patients’ motivation and expectations of robot-assisted gait training (RAGT), and therapists’ perspectives on the usability of RAGT	IMI	Once
Prange GB [49]	2015	RCT	Netherlands	SA	To examine the effect of weight-supported arm training combined with computerized exercises on arm function and capacity, compared with dose-matched conventional reach training	IMI	Once (post-training)
Johar MN [50]	2022	RCT	Malaysia	C	To assess the effectiveness of game-based circuit exercise in comparison to conventional circuit exercise on functional outcome (lower limb strength, postural stability, and aerobic endurance), motivation level, self-efficacy, and quality of life.	IMI	Baseline, after 12 and 24 weeks
Hung NT [51]	2021	CT	USA (Illinois)	C	To assess tolerability and feasibility of home-based, high-dose “myoelectric interface for neurorehabilitation training” therapy	IMI	At the end of the 6th training week
Thielbar KO [52]	2020	RCT	USA (Illinois)	C	To compare participation and subjective experience of participants in both home-based multiuser virtual reality (VR) therapy and home-based single-user (SU) VR therapy	IMI	2 and 4 weeks after beginning of the intervention
Nijenhuis SM [53]	2015	CT	Netherlands, United Kingdom, Italy	C	To assess the feasibility and potential clinical changes associated with a technology-supported arm and hand training system at home for patients with chronic stroke	IMI	After intervention
Nijenhuis SM [54]	2017	RCT	Netherlands	C	To compare user acceptance and arm and hand function changes after technology-supported training at home with conventional exercises	IMI	After intervention
Subramanim S [55]	2014	CT	USA (Illinois)	C	To assess adherence and intervention-related effects of gaming to improve balance control and decrease cognitive-motor interference	IMI	Pre- and post- intervention
Friedman N [56]	2014	RCT	USA (California)	C	To compare the effect of training with a specific robotic glove to conventional hand therapy	IMI	After each session
Lloréns R [57]	2015	RCT	Spain	C	To evaluate the clinical effectiveness of a virtual reality (VR)-based telerehabilitation program.	IMI	After treatment
Knippenberg E [58]	2021	CT	Belgium	SA + C	To evaluate the usability, credibility, and treatment expectancy of i-ACT (intelligent activity-based client-centred task-oriented training) and the motivation towards i-ACT use in rehabilitation over time	IMI	At baseline or after one training session with i-ACT, after 2 weeks, 4 weeks, and 6 weeks of training, and 9 weeks after the cessation of training
Popović MD [59]	2014	RCT	Serbia	SA	To investigate whether feedback-mediated exercise (FME) of the affected arm of hemiplegic patients increases patient motivation and promotes greater improvement of motor function, compared to no-feedback exercise	IMI	Baseline and after three weeks of treatment
Novak D [60]	2014	RCT	Switzerland	C	To explore the potential of two-player game, played using two robotic devices designed for arm rehabilitation. Three game modes were tested: single-player (competing against computer), competitive (competing against human), and cooperative (cooperating with human against computer)	IMI	After each game mode
Chen L [61]	2019	RCT	China	DS	To compare the efficacy of motor relearning program versus Bobath approach	AES	After 1, 3, 6, 9, and 12 months
Radder B [62]	2018	CT	The Netherlands	C	To investigate the feasibility of a wearable, soft-robotic glove system developed to combine assistive support in daily life with performing therapeutic exercises on a computer at home	IMI	Once (at end of Phase 2)
Ahmad MA [63]	2019	RCT	Malaysia	C	To examine the effectiveness of VR games as an adjunct to standard physiotherapy in improving upper limb (UL) function and general health.	IMI	Before and after 8 interventional weeks
Rapolienė J [64]	2018	RCT	Lithuania	A	To evaluate the effectiveness of occupational therapy on motivation.	MHLC	Before first occupational treatment and after 20 occupational therapy procedures
Lin FH [65]	2019	RCT	Taiwan	C	To investigate the effects of routine rehabilitation activities and additional social support and health education by functional therapists on motivation and post-stroke depression	CES-D10	At the 2nd week, 4th week, 8th week, and 4 weeks after the end of the study
Hung JW [66]	2016	CT	USA (Illinois)	C	To investigate the feasibility and potential efficacy of the video-controlled biofeedback system for balance training	IMI	At the end of the 6-week training
LaPiana N [67]	2020	CT	USA Washington	A	To assess the acceptability of a smartphone-based augmented reality game	Ad hoc developed questionnaire	At the end of the final gaming session
Huber SK [68]	2021	CT	Switzerland	C	To investigate the feasibility of a rehabilitation approach using user-centered exergames	BREQ-3 and the SMSC	At baseline and after 8 weeks
Graven C [69]	2011	RCT	Australia	DS	To assess the efficacy of an integrated approach (including written provision at discharge, additional phone contacts, and home visits) to facilitate patient goal achievement in the first year post-stroke	GDS-15	At baseline and six and twelve months post-stroke
Morice E [70]	2020	RCT	Switzerland	SA	To assess the effects of dance program on patients’ balance control, cognitive function, strength, coordination, functional status, balance confidence, quality of life, and motivation	MAPS	Before and after eight weeks of intervention
An HJ [71]	2021	RCT	Republic of Korea	C	To investigate the effect of animal-assisted therapy on gait performance, pulmonary function, and psychological variables	SRMS	Before and after eight weeks of intervention
Adhikari SP [72]	2018	RCT	Nepal	SA	To examine the immediate effect of “action-observation-execution” with accelerated skill acquisition program (ASAP) on dexterity	SRMS	At baseline, after training, and during one-week follow-up
Thompson N [73]	2022	CT	United Kingdom	SA	To assess the feasibility and acceptability of delivering neurologic music therapy one day-per-week	Ad hoc developed questionnaire	Before and after each single session
Street AJ [74]	2018	RCT	United Kingdom	SA + C	To evaluate music therapy as a home-based intervention for arm hemiparesis	Ad hoc developed questionnaire	After 1, 6, 9, and 15 weeks
Morone G [75]	2015	CT	Italy	DS	To evaluate the feasibility of brain–computer interface-assisted motor imagery training to support hand/arm motor rehabilitation	QCM	At the end of each training session
Deguchi K [76]	2013	RCT	Japan	DS	To assess the usefulness of a novel computerized touch panel-type screening test	AES -VI	Unknown
Seregni A [77]	2021	CT	Italy, Spain	C	To assess the efficacy of a novel virtual coach system in assisting and counselling patients during home rehabilitation activities	Ad hoc developed questionnaire	Unknown
Chen HM [78]	2020	CT	Taiwan	A + SA	To determine whether motivational interviewing (MI) improves the performance of activities of daily living and enhances motivation for rehabilitation among first-stroke patients	RMS	Baseline, 6 weeks, and three months after the intervention
Aramaki AL [79]	2019	CT	Brasil	SA + C	To analyse the feasibility of a rehabilitation protocol using client-centered VR and to evaluate changes in occupational performance and social participation	PS	Baseline and after treatment (12 weeks)
Wissink KS [80]	2014	CT	The Netherlands	C	To determine therapy intensity of and motivation for physical therapy of geriatric stroke patients in nursing homes, its correlates, and the effect on discharge destination	PRPS	During three interventional weeks
Chowdhury A [81]	2015	RCT	India	C	To assess the validity of a rehabilitation protocol, characterized by a combination of mental practice (MP) and physical practice (PP), using a hand exoskeleton and brain-computer interface (BCI)	VAS scale	After each session
Song HS [82]	2019	RCT	Republic of Korea	C	To determine the difference in self-satisfaction by comparing class-based task-oriented circuit training and individual-based task-oriented circuit training	MRS	Pre- and post-intervention
Park JS [83]	2019	RCT	Republic of Korea	SA	To investigate the effect of game-based exercise on hand strength, motor function, and compliance	VAS scale	After every training session
Alhirsan SM [84]	2021	RCT	USA (Alabama)	C	To demonstrate the different effects of augmented feedback, simple VR, and exergaming applications on motivation and walking speed performance	IMI	After each condition
Grau-Sánchez J [85]	2021	RCT	Spain	C	To assess the efficacy of enriched music-supported therapy on cognitive functions, emotional well-being, and quality of life	IMI + AES	IMI: during intervention; AES: baseline, pre-, and post-intervention
Zhang L [86]	2022	RCT	China	C	To explore whether coaching-based teleoccupational guidance will help stroke survivors and caregivers to obtain satisfactory outcomes	IMI	Baseline and after 3 and 6 months
Rozevink SG [87]	2021	RCT	The Netherlands	C	To investigate the effect of robotic device combined with a telecare platform on the upper limb function of patients with unilateral upper limb paresis	IMI	Once (post-treatment)
Skidmore ER. [88]	2015	RCT	USA (Pennsyl-vania)	A	To examine the effects of a behavioural intervention, used to augment usual inpatient rehabilitation, on apathy symptoms	AES	Baseline and after 3 and 6 months
Cano-Mañas MJ [89]	2020	RCT	Spain	SA	To determine the effect of a structured protocol using commercial video games on balance, postural control, functionality, quality of life, and level of motivation.	Ad hoc developed questionnaire	Pre- and post-treatment

Abbreviations: Study design—CT: clinical trial; RCT: randomized clinical trial. Stroke setting—A: acute; SA: subacute; C: chronic; DS: different stages. Motivation assessment tool—IMI: Intrinsic Motivation Inventory; AES: Apathy Evaluation Scale; MHLC: Multidimensional Health Locus of Control scale; CES-D10: 10-item Center for the Epidemiological Studies of Depression Short Form; BREQ-3: Behavioural Regulation in Exercise Questionnaire; SMSC: Sport- and Movement-Specific Self-Concordance Scale; GDS-15: Geriatric Depression Scale—15 items; MAPS: echelle de Motivation envers l’Activité Physique en contexte de Santé; SRMS: Stroke Rehabilitation Motivation Scale; QCM: Questionnaire for Current Motivation; VI: Vitality Index; RMS: Rehabilitation Motivation Scale; PS: Participation Scale; MRS: Motivation of Rehabilitation Scale; VAS: Visual Analogic Scale; PRPS: Pittsburgh Rehabilitation Participation Scale.

**Table 3 ijerph-20-04187-t003:** Specifically designed for stroke patients in rehabilitation motivation scales.

	AAMQ	MORE	28-SRMS
First Author	Derakh-shanrad SA [94]	Yoshida T [93]	White GN [92]
Year	2016	2022	2012
Questionnaire was developed from	The Persian version of Hermans Achievement Motivation Questionnaire (PHAMQ)	An item pool, created by nine rehabilitation professionals	Sports Motivation Scale (SMS)
Number of enrolled patients in the scale development	25	201	18
Number of items	28	17	28
Response format	4-point scale	7-point scale	5-point scale
Internal consistency (Cronbach’s alpha coefficient)	0.946	0.948	0.5
Validity	Content validity was approved by an expert panel. Criterion validity could not be assessed because there was no regionally standardized comparison measure. A factorial analysis (construct validity) could not be performed due to the small sample size.	Construct validity was investigated using exploratory factor analysis, confirmatory factor analysis, and item response theory analysis. Criterion validity was investigated through Spearman’s rho, calculated between the MORE scale and the Apathy Scale (AS), Self-rating Depression Scale (SDS), and Visual Analogue Scale (VAS).	The scale’s validity could not be proved in the original version.
Notes (questionnaire dimensions, subscales, etc.)	Although a factorial analysis was not performed, the authors suggest the possibility of latent construct of the scale, which is based on the following factors: perseverance, self-esteem, time perception, seeking opportunities, diligence, competency, high ambition, and foresight.	The factor analysis suggests a one-factor structure (it was impossible to separate the motivation-related factors from the relevant item).	7 factors have been identified: EM-introjected; EM-regulation; EM-identification; amotivation; IM-knowledge; IM-stimulation; IM-accomplishment.

Abbreviations: AAMQ: Adapted Achievement Motivation Questionnaire; SRMS: Stroke Rehabilitation Motivation Scale; MORE: Motivation for Rehabilitation scale; EM: extrinsic motivation; IM: intrinsic motivation.

## Data Availability

No new data were created or analyzed in this study. Data sharing is not applicable to this article.

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
