# Peer review of "Motivation as a Measurable Outcome in Stroke Rehabilitation: A Systematic Review of the Literature"

_ijerph, 2023, doi:10.3390/ijerph20054187_

Round 1

Reviewer 1 Report

Reading the introduction part carefully, I found that the article is very well founded from a theoretical point of view, but still, I have some recommendations:

·        Lines 47-48: Research on motivation has been employed in multiple areas, including applications in business, educational, wellness fields.   The recommendation is to insert a few citations covering the three areas.

·        Lines 53-54: Despite these observations, in the rehabilitation literature exists nowadays a lack of consensus on nature, clinical measurement and determinants of motivation [4].   Since 2000 (when the article quoted was published) and until now, things have changed a lot.  The recommendation is that that sentence is removed from the article or another sentence be inserted with a different citation.

1.      Lines 186-187: A literature search (PubMed and Google Scholar) was performed from 1st January 2010 to 30th Juny 2022.  Please specify exactly when the procedure for searching the articles was started and the criteria for their inclusion (2010-30th Juny 2022).

2.      Line 231: The 46 selected studies were published between 2010 and 2022.  The 46 selected studies were published between 2011 and 2022.  Their inclusion criterion was the 2010-2022 timeframe.

3.      Lines 277-278: Furthermore, in some studies of our review, motivation has been also indirectly quantified by depression or participation related assessments.  Which were these studies you are discussing?

4.      Lines 353-354: In a validation study of the EMAP in Spanish* [97], the factor analysis confirmed the original 6 dimensions of EMAP. *Spanish Population

5.      The recommendation is that the conclusions should be shorter, to the point, concerning the results obtained.

6.      The article was included to analyse the similarity coefficient in the Plagiarism CheckerX software, and please perform the following word reformulations that are in bold and italics:

·        Five central aspects of patient participation (Respect and integrity, Planning and decision making, Information and knowledge, Motivation and encouragement, and Involvement of family) are covered.

Author Response

Dear reviewer,

we are very grateful for your comments. According to your suggestions, we have revised the manuscript extensively. Our responses are given in a point-by-point manner below.

We hope that we could clarify all outstanding issues to your full satisfaction.

Sincerely,

Giulio Verrienti

----------------------------------------------------------------------------------------

  • Lines 47-48: Research on motivation has been employed in multiple areas, including applications in business, educational, wellness fields.   The recommendation is to insert a few citations covering the three areas.

Response: We thank the reviewer for this suggestion and inserted the following citations:

Business:

  • Deci, E. L., Olafsen, A. H., & Ryan, R. M. (2017). Self-determination theory in work organizations: The state of a science. Annual Review of Organizational Psychology and Organizational Behavior, 4, 19–43. org/10.1146/annurev-orgpsych-032516-113108
  • Steers, R. M., Mowday, R. T., & Shapiro, D. L. (2004). Introduction to special topic forum: The future of work motivation theory. The Academy of Management Review, 29(3), 379–387. org/10.2307/20159049
  • Latham, G. P. (2007). Work motivation: History, theory, research, and practice. Sage Publications, Inc.

Educational:

  • Tohidi, Hamid. (2012). The effects of motivation in education. Procedia - Social and Behavioral Sciences. 31. 820 – 824. 10.1016/j.sbspro.2011.12.148.
  • Stirling, D. (2013). Motivation in Education. Aichi Universities English Education Research Journal. 29. 51-72.

Wellness

  • Gillison F.B, Rouse P., Standage M., Simon J. Sebire & Richard M. Ryan (2019) A meta-analysis of techniques to promote motivation for health behaviour change from a self-determination theory perspective, Health Psychology Review, 13:1, 110-130, DOI: 10.1080/17437199.2018.1534071

  • Lines 53-54: Despite these observations, in the rehabilitation literature exists nowadays a lack of consensus on nature, clinical measurement and determinants of motivation [4].   Since 2000 (when the article quoted was published) and until now, things have changed a lot.  The recommendation is that that sentence is removed from the article or another sentence be inserted with a different citation.

Response: Based on this comment, we removed this sentence from the text.

  1. Lines 186-187: A literature search (PubMed and Google Scholar) was performed from 1st January 2010 to 30th Juny 2022.  Please specify exactly when the procedure for searching the articles was started and the criteria for their inclusion (2010-30th Juny 2022).

Response: We thank the reviewer for this important recommendation. We now specify when the procedure of searching the articles was started and the criteria for their inclusion.

A literature search (PubMed and Google Scholar) was performed on 22nd August 2022 and we selected articles published in the timeframe from 1st January 2010 to 30th Juny 2022, using the following Medical Subject Headings (MeSH) terms: “Assessment” OR “Scale” AND “motivation” AND “stroke” AND “rehabilitationIn addition, a backward search (checking bibliography of identified papers) was conducted to identify any studies that were not retrieved using the main search strategy. The inclusion criteria were (1) articles published in English language; (2) interventional studies in stroke patients in rehabilitation setting, which examine the direct effect of the intervention on motivation, measured by specifically designed assessment tools; (3) original research examining the indirect effect of an intervention on motivation (in this case the motivating effect of the intervention is measured by indirect motivation assessment tools, such those specifically designed to assess participation or depression); and (4) studies conducted in the above reported period. In the main search strategy, following exclusion criteria were adopted: (1) Article-types such as letters to the editor, case reports, reviews, meta-analyses (2) studies for which the complete text could not be found (3) articles not in English.  Two authors independently performed all searches and removed duplicate records. Finally, a third author performed a quality assessment. “

  1. Line 231: The 46 selected studies were published between 2010 and 2022.  The 46 selected studies were published between 2011 and 2022.  Their inclusion criterion was the 2010-2022 timeframe.

Response: We thank the reviewer for this comment.      We added the timeframe as an inclusion criterion accordingly.

  1. Lines 277-278: Furthermore, in some studies of our review, motivation has been also indirectly quantified by depression or participation related assessments.  Which were these studies you are discussing?

Response: We thank the reviewer for this point. We now added the citations we referred to, as follows:

(61) Chen L, Xiong S, Liu Y, Lin M, Zhu L, Zhong R, Zhao J, Liu W, Wang J, Shang X. Comparison of Motor Relearning Program versus Bobath Approach for Prevention of Poststroke Apathy: A Randomized Controlled Trial. J Stroke Cerebrovasc Dis. 2019 Mar;28(3):655-664. Doi: 10.1016/j.jstrokecerebrovasdis.2018.11.011. Epub 2018 Nov 28. PMID: 30501977.

(65) Lin FH, Yih DN, Shih FM, Chu CM. Effect of social support and health education on depression scale scores of chronic stroke patients. Medicine (Baltimore). 2019 Nov;98(44):e17667. Doi: 10.1097/MD.0000000000017667. PMID: 31689780; PMCID: PMC6946326.

(69) Graven C, Brock K, Hill K, Ames D, Cotton S, Joubert L. From rehabilitation to recovery: protocol for a randomised controlled trial evaluating a goal-based intervention to reduce depression and facilitate participation post-stroke. BMC Neurol. 2011 Jun 18;11:73. Doi: 10.1186/1471-2377-11-73. PMID: 21682910; PMCID: PMC3135526.

(76) Deguchi K, Kono S, Deguchi S, Morimoto N, Kurata T, Ikeda Y, Abe K. A novel useful tool of computerized touch panel-type screening test for evaluating cognitive function of chronic ischemic stroke patients. J Stroke Cerebrovasc Dis. 2013 Oct;22(7):e197-206. Doi: 10.1016/j.jstrokecerebrovasdis.2012.11.011. Epub 2013 Jan 2. PMID: 23290436.

(79) Aramaki AL, Sampaio RF, Cavalcanti A, Dutra FCMSE. Use of client-centered virtual reality in rehabilitation after stroke: a feasibility study. Arq Neuropsiquiatr. 2019 Sep 23;77(9):622-631. Doi: 10.1590/0004-282X20190103. PMID: 31553392

(80) Wissink KS, Spruit-van Eijk M, Buijck BI, Koopmans RT, Zuidema SU. CVA-revalidatie in het verpleeghuis: therapie-intensiteit van en motivatie voor fysiotherapie [Stroke rehabilitation in nursing homes: intensity of and motivation for physiotherapy]. Tijdschr Gerontol Geriatr. 2014 Jun;45(3):144-53. Dutch. doi: 10.1007/s12439-014-0072-6. PMID: 24801121

(85) Grau-Sánchez J, Segura E, Sanchez-Pinsach D, Raghavan P, Münte TF, Palumbo AM, Turry A, Duarte E, Särkämö T, Cerquides J, Arcos JL, Rodríguez-Fornells A. Enriched Music-supported Therapy for chronic stroke patients: a study protocol of a randomised controlled trial. BMC Neurol. 2021 Jan 12;21(1):19. Doi: 10.1186/s12883-020-02019-1. PMID: 33435919; PMCID: PMC7801568.

(88) Skidmore ER, Whyte EM, Butters MA, Terhorst L, Reynolds CF 3rd. Strategy Training During Inpatient Rehabilitation May Prevent Apathy Symptoms After Acute Stroke. PM R. 2015 Jun;7(6):562-70. Doi: 10.1016/j.pmrj.2014.12.010. Epub 2015 Jan 13. PMID: 25595665; PMCID: PMC4466065.

  1. Lines 353-354: In a validation study of the EMAP in Spanish* [97], the factor analysis confirmed the original 6 dimensions of EMAP. *Spanish Population

Response: we corrected this wording accordingly

  1. The recommendation is that the conclusions should be shorter, to the point, concerning the results obtained.

Response: Based on the reviewer´s suggestion we shortened the conclusion and focused on the results described in the article.

  1. The article was included to analyse the similarity coefficient in the Plagiarism CheckerX software, and please perform the following word reformulations that are in bold and italics:
  • Five central aspects of patient participation (Respect and integrity, Planning and decision making, Information and knowledge, Motivation and encouragement, and Involvement of family) are covered.

Response: Based on the reviewer´s suggestion we corrected the sentence accordingly

Reviewer 2 Report

The presented paper presents an overview and analysis of the assessment of motivation in patients with stroke. The article is of great interest to clinical psychologists and rehabilitation specialists. Complete conclusions on this issue are presented.

I have only one remark:

It is known that the age of the patient can also play a significant role in the rehabilitation process. This fact is reflected even in approaches to the rehabilitation of such patients. Perhaps the authors should also mention this, and if there is an opportunity to evaluate this factor from the articles submitted for analysis, if there is such an opportunity.

Author Response

Dear reviewer,

Thank you for your positive comments on our manuscript. We agree that age is a determining factor in the rehabilitation process. Furthermore, it is reasonable that younger patients are more motivated than older. However, to our knowledge, how age shows its influence on motivation should be further cleared. Unfortunately, we have not performed a specific analysis on this topic, because the included studies are too different to perform it. Anyways, we underlined age as a motivation-determining factor and we mentioned it as a possible adjustment factor in further research direction (see lines 440-442).

We hope that we could clarify all outstanding issues to your full satisfaction.

Sincerely,

Giulio Verrienti

Round 2

Reviewer 1 Report

Congratulations on your work in conducting this systematic review.